# Signature of gravity wave propagations from the Troposphere to Ionosphere

Hisao Takahashi[1], Cosme A. O. B. Figueiredo[1], Patrick Essien[2], Cristiano M. Wrasse[1], Diego Barros[1], Prosper K. Nyassor[1], Igo Paulino[3], Fabio Egito[3], Geangelo M. Rosa[4], Antonio H. R. Sampaio[4]

1. Space Weather Division, Instituto Nacional de Pesquisas Espaciais, São José dos Campos, Brazil

2. University of Cape Coast, Department of Physics, Cape Coast, Ghana

3. Unidade acadêmica de Física, Universidade Federal de Campina Grande, Campina Grande, Brazil

4. Federal Institute for Education, Science and Technology Baiano (IF Baiano), Bom Jesus da Lapa, Brazil

*Correspondence to*: Hisao Takahashi (hisao.takahashi@inpe.br)

**Abstract.** We observed a gravity wave (GW) signature in the OH emission layer in the upper mesosphere, and 4 hours later, a medium-scale traveling ionospheric disturbance (MSTID) in the OI 630 nm emission layer. Spectral analysis of the two waves did show that both have almost the same wave characteristics: wavelength, period, phase speed and propagation direction, respectively, 200 km, 60 min, 50 m/s, propagating toward the Southeast. During the MSTID occurrence, concentric wavefronts were also observed in the ionosphere by detrended total electron content (dTEC) maps. From the gravity wave ray-tracing simulation for the mesospheric gravity wave, we found that the wave came from a tropospheric deep convection spot and propagated up to the 140 km altitude. Regarding the same wave characteristics between mesospheric GW and MSTID, two possible cases are investigated: a direct influence of the GW oscillation in the OI 630 nm emission height and the generation of a secondary wave during the GW breaking process. The concentric wave structure suggests the generation of a secondary wave after the primary wave was dissipated in the lower thermosphere. This is the first time to report an observational event of gravity wave propagation from the troposphere, mesosphere to thermosphere-ionosphere in the south American region.

## 1. Introduction

A deep cloud convection in the troposphere generates vertical (up and down) air-mass movement launching a variety of gravity waves into the stratosphere. Atmospheric gravity waves (GWs) have important roles in transporting the energy and momentum from the lower to upper atmosphere and ionosphere. A part of energy and momentum is deposited in the mesosphere lower-thermosphere (MLT) region through wave breaking and altering the background wind field. Some of the GWs produce secondary waves and propagate further upwards into the thermosphere where it modulates ionospheric plasma (Hocke and

Schlegel,1996, Nicolls et al., 2014). A part of Medium-Scale Travelling Ionospheric Disturbances (MSTIDs) has its origin in

the passage of gravity waves in the ionosphere (Otsuka, 2018). Observations of GW propagation in the thermosphere have been carried out by many researchers since Hines (1960) has presented theoretical background for the GW propagation in the ionosphere. Rottger (1973) suggested the role of GWs in the ionospheric irregularities.

GW observations in the mesosphere have been carried out by measuring short period temporal variation of the mesospheric
airglow (Hydroxyl and atomic oxygen OI 557.7 nm emissions) by airglow photometers in 1970-1990 (e. g., Takahashi et al, 1999). After 1990 airglow digital imagers were used to monitor GWs in two-dimensional forms (e.g., Taylor et al. 2009, Dare-Idowu et al. 2020, Nyassor et al., 2021). In case of GWs in the stratosphere, satellite-onboard GPS radio occultation measurements have made it possible to observe GWs by vertical profile of the temperature variability on a global scale (Tsuda, 2014, Xu et al., 2017).

There are many previous works on the GW propagations in the stratosphere, mesosphere and ionosphere individually. However, it has been difficult to monitor an event of GW propagating through the troposphere up to the ionosphere. Smith et al. [2013] observed GW waves in the OH and OI 557.7 nm emission layers in the mesosphere to lower thermosphere (MLT) region (85-100 km) and OI 630.0 nm emission layer (around 240 km altitude) in the thermosphere and discussed on the mountain waves from the mesosphere to ionosphere. They attributed the wave structure in the ionosphere as due to secondary
waves. Azeem et al. (2015), for the first time, reported the occurrence of circular GW structures in the stratosphere, mesosphere and ionosphere during a tropospheric convective storm. They observed concentric wave structures in the stratosphere by the Atmospheric Infrared Sounder(AIRS) onboard Aqua satellite (https://airs.jpl.nasa.gov/), and by an optical imaging radiometer (VIIRS) onboard Suomi satellite (https://www.nasa.gov/mission_pages/NPP/main/index.html), and in the ionosphere by ground-based GPS receivers. Prior to this work, Nishioka et al. (2013) has reported concentric gravity waves in the ionosphere
which were induced by a severe convective system (supercell) in the troposphere. The concentric waves lasted for more than 7 hours. Nyassor et al. (2021) reported the first mesospheric concentric gravity waves excited by thunderstorm. Takahashi et al. (2020) presented the generation and propagation of MLT-GWs and concentric MSTIDs in the ionosphere during a deep convection activity in the troposphere over the south American continent.

Regarding propagation of GWs from the lower to upper atmosphere, Vadas (2007), as the first time, studied propagation
property of GWs from the troposphere to the thermosphere for the horizontal wavelength of 10 to 1000 km and the period of 10 to 100 min. The author presented the GW dissipation altitudes depending on their horizontal wavelength and period. In case of the horizontal wavelength of 200 km and its period of 60 min, for example, the model predicts dissipation above 120 km altitude. It means that the dissipation produces a body force and generates secondary waves.

There is a difficulty to observe a gravity wave from its origin (source) in the troposphere following up to the thermosphere.
During the upward propagation, it could change its wave characteristics under the background atmosphere condition, dissipating and producing secondary waves changing the horizontal wavelength, phase speed and propagation direction (Vadas and Crowley, 2010). It would take several hours to reach from the troposphere to the mesosphere-lower thermosphere (Vadas

and Liu, 2013), which makes it difficult to follow the wave step by step. Recent observation of concentric wavefronts in the stratosphere, mesosphere and thermosphere (Azeem et al, 2015) would be rather a rare case. Further observational evidence would be necessary to clarify the propagation processes. The purpose of the present work is to report a case of gravity wave propagation directly from Tropospheric convection to the mesosphere and thermosphere/ionosphere. For investigating the propagation of gravity waves, data from airglow OH imager in the mesosphere and OI 630 nm imager from the thermosphere/ionosphere, ionosondes and a GNSS receiver network are used.

### 2. Observations

Airglow observation has been carried out at Bom Jesus da Lapa (hereafter BJL), 13.3° S, 43.5° W, geomag.14.1° S, since 2019. The observation site is located under the equatorial ionospheric anomaly (EIA) belt. Equatorial plasma bubbles can also be frequently observed. An all-sky airglow imager equipped with 3-inch optical interference filters (for 630.0 nm, 557.7 nm and OH-NIR(710-930 nm) takes 180° wide images with a time sequence of ~ 5 min. Exposure time for each filter is 15 s for the OH-NIR and 90 s for the OI 630.0 nm and 557.7 nm images. The imager characteristics have been presented by Wrasse et al. (2021). In the present study we used the image data from December 2019 to September 2020. During this period, we selected 13 days of observation to analyze wave structures in the OI 630 nm images.

Ionospheric Total Electron Content (TEC) maps are produced by using ground-based GNSS receiver network over the South America (Takahashi et al., 2016). The TEC maps over South America with 10 min intervals are available at EMBRACE's website (http://www2.inpe.br/climaespacial/portal/en/). To retrieve small amplitude of TEC variability, less than 1.0 TEC unit of spatial variation, we plot the vertical TEC time series from the individual receiver, and to pick up a fraction from running means as follows,

dTEC (t) = TEC(t) - <TEC(t +/- 30 min)>

where the symbol < > means the time average between the +/- 30 min interval. The temporal resolution of the dTEC map is one min. The spatial resolution is a square of 0.25° x 0.25°, then taking a running average with 5 points. The dTEC(t), therefore, can be plotted on the map with 1.25° square (~130x130 km) by one min time interval. Any wave propagation mode, of which oscillation amplitude being less than one TECu and the temporal oscillation between 5 and 30 min can be plotted (Figueiredo et al. 2018, Essien et al. 2021).

In the present work, the data from the ionosondes were used to observe the vertical drift of the F-layer and to calculate the electron density profile. Three digital ionosondes (DPS-4) (http://www.digisonde.com/instrument-description.html) have been operated one at São Luís (2.6° S, 44.2° W, geomag. 3.9°S), Fortaleza (3.9° S, 38.4° W, geomag. 6° S), and Cachoeira Paulista (22.7° S, 45.0° W, geomag. 18.1° S). The DPS-4 sounder has a 500 W peak power, covering a frequency range from 0.5 to 30 MHz. Ionograms are taken with a time interval of 10 minutes.

### 3 Results

## 3.1 OH images

On the night of 18-19 January 2020, around 22:30 to 23:30 UT (19:30-20:30 Local Time), the airglow OH images showed two wave structures passed over BJL. Figure 1 shows the OH images at the moment of one of the wavefronts passing over the zenith. The images are projected on the geographic coordinates. The horizontal extension of the image is approximately 500 km and the blue dot indicates the location of the BJL observation site. In Figure 1(a), there are two wave structures, one is shorter wavelength in the northwest side of the image (top left side) (GW-1) and another is a longer wavelength, one wavefront

over the zenith (blue dot) and the other at the SE, indicated by the blue arrows (GW-2). Three sequential images with a time interval of 10 minutes indicate that the wavefront is moving toward the SE (indicated by a red arrow in Fig 1(b)). The propagation mode can be clearly seen if keograms are made by using the meridional and longitudinal cuts as a function of time. Figure 2 shows the keograms (zonal and meridional cuts) between 22:30 and 28:30 (04:30) UT. Looking at the GW-2, two bright bands propagating from the North to South and the West to East can be seen. The broad bright band passing over

the zenith from the NE to SW through the night is the Galactic Milky way.

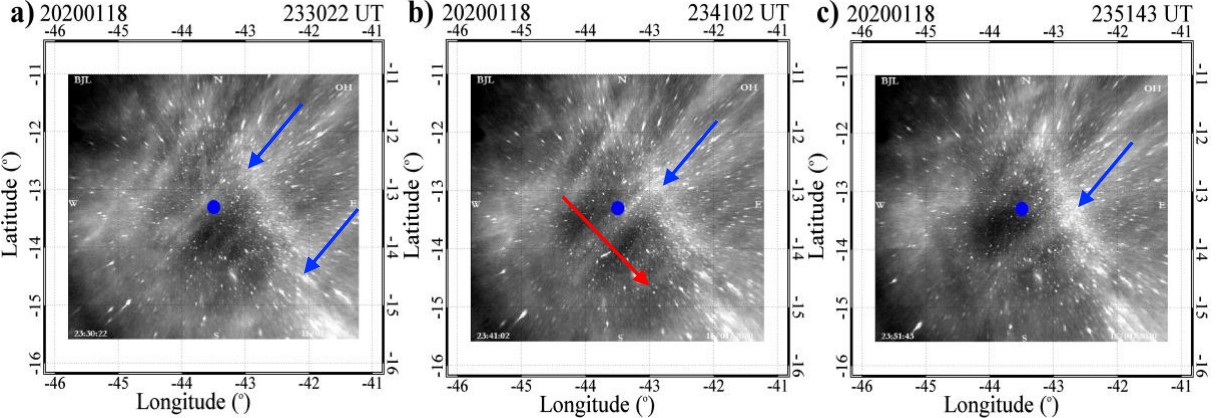

*Fig. 1. Geographically coordinated OH images observed at Bom Jesus da Lapa (BJL), 13.3° S, 43.5° W, geomag.14.1° S, at 23:30 UT (left), 23:41 UT (center) and 23:51 UT (right) on the night of 18-19 January 2020. Blue dots indicate the zenith of BJL. The blue arrows indicate the wave fronts of the longer wavelength one (GW-2). The red-arrow indicates the direction of propagation.*

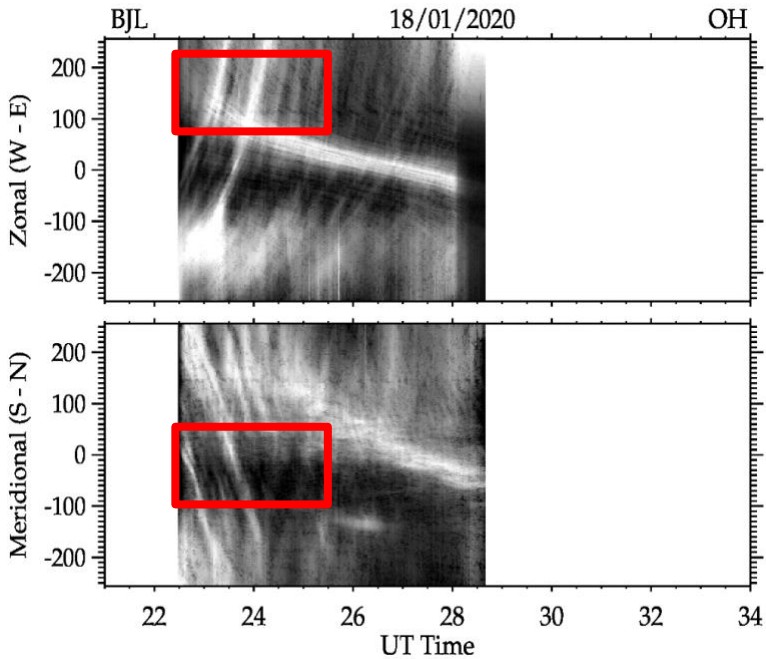

*Fig. 2. Keograms of OH images observed at BJL in the night of 18-19 January 2020. Meridian cuts along the W-E direction (upper panel) and the S-N direction (lower panel) are shown as a function of time from 22:00 to 30:00 (06:00) UT. The red color rectangular boxes are where the FFT spectral analysis were taken.*

For calculating the wave characteristics (horizontal wavelength, period, phase velocity), we used Fast Fourier Transform (FFT) spectral analysis (Wrasse et al., 2007, Figueiredo et al., 2018, Essien et al., 2018). Image samples used in the calculation are indicated by red boxes in Figure 2. The wave characteristics of the longer wave (GW-2) are, the horizontal wavelength of 217.9±12 km, the period of 60.6±03 min, the phase speed of 59.9±5 m/s, and the propagation direction of 148.5± 10°. For the short wave (GW-1) the wave characteristics was also obtained: the horizontal wavelength of 36.2±1 km, the period of 15.8±0.8 min, the phase speed of 38.3±2 m/s, and the propagation direction of 135.0±10°.

### 3.2  OI 630 nm image

Figure 3 presents 3 sequential images of the OI 630 nm emission between 03:20 UT and 03:48 UT.  The top 3 panels are original images, and the bottom 3 panels are the residual images which are subtracted from the one hour averaged image. From the residual images one can see two dark bands in the southwest of BJL propagating toward East, which seem to be the Medium Scale Travelling Ionospheric Disturbance, named as MSTID-1. We checked any contamination of the OH emission in the OI

630 nm image. No such wavelike structure could be observed in the OH images during the same period. The bright OI 630 emission intensity over the northwest part of the sky should be the midnight downward drift of the F-layer accompanied by the Midnight Temperature Maximum (MTM) in the thermosphere (Colerico et al.,1996, Figueiredo et al., 2017). One can also notice the presence of the equatorial plasma bubbles (EPBs) (at least two depletions) in the northwest of BJL, which are also

drifting toward the East. The difference between the EPBs and MSTID-1 is clear to see. The EPBs are extending from the equator side and the MSTID is elongating from the South. During the 28 min of the time interval (from Figure 3(a) to (c), a dark band moved toward the East by ~90 km. In order to get the wave characteristics of MSTID-1 from the OI 630 images, we used the FFT spectral analysis for the OI 630 keogram (not shown here), which is similar to the OH image analysis mentioned above (Wrasse et al, 2007). The results are the horizontal wavelength of 201.7±13 km, the period of 64.2 ± 33 min,

the phase speed of 52.4 ± 27m/s and the propagation direction of 113.2 ±10°. The characteristics of the wave propagation in the OH emission layer and OI 630 nm emission layer are summarized in Table 1. The movement of wave fronts of MSTID-1 is presented in the supporting file: [OI 6300_movie.mp4].

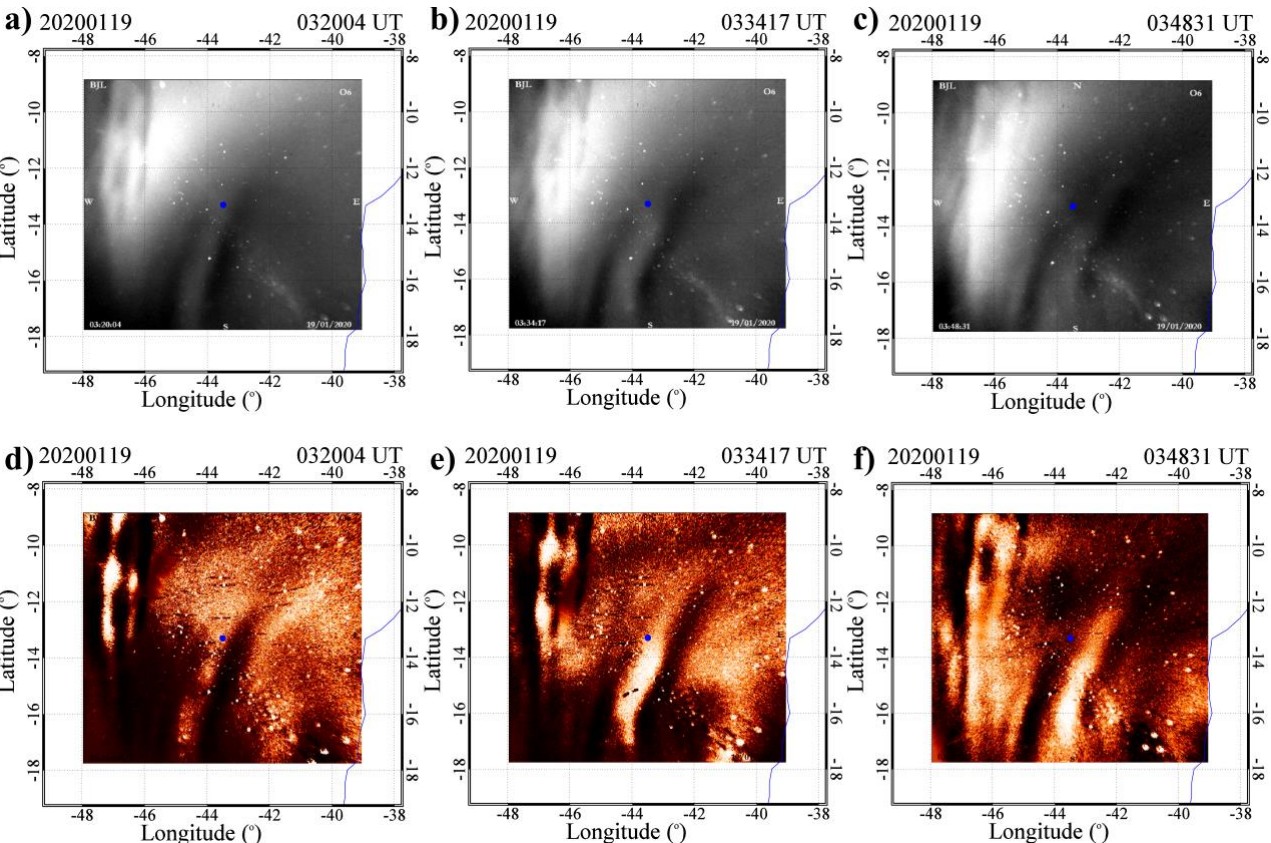

*Fig. 3. Geographically coordinated OI 630 nm images observed at Bom Jesus da Lapa (BJL) at 03:20 UT (left), 03:34 UT(center) and 03:48 UT (right) on the night of 18-19 January 2020. The top 3 images are original and the bottom 3 images*

*are residual subtracted from the one hour averaged image. Blue dots indicate the location of BJL. MSTID-1 (lower/middle),*
*EPBs (upper left corner) and MTM (upper left) can be seen (see Text).*


***Table 1.*** *Wave Characteristics obtained by the OH images in the MLT region (GW-1 and GW-2), and OI 630 nm*
*images in the thermosphere (MSTID-1) and dTEC maps in the ionosphere (MSTID-2). (The values in parentheses*
*are the error range.)*

| GWs & MSTIDs | OH (22-00 UT) (GW-1) | OH (23-00 UT) (GW-2) | OI630 (03-04 UT) (MSTID-1) | dTEC (02-04 UT) (MSTID-2) |
|---|---|---|---|---|
| $\lambda_H$ (horiz wave length) (km) | 36.2 (1.0) | 217.9 (12.3) | 201.7 (13.4) | 360 (50) |
| $\tau$ (period) (min) | 15.8 (0.8) | 60.6 (3.0) | 64.2 (33.2) | 26.5 (2) |
| Vp (phase speed) (m/s) | 38.3 (2.0) | 59.9 (4.5) | 52.4 (26.9) | 226 (20) |
| Az (Azimuth) (deg) | 135.0 (10.0) | 148.5 (10.0) | 113.2 (10.0) | Circular |


### 3.3  dTEC map

A small amplitude of perturbation of the ionospheric total electron content (TEC) was retrieved from the dTEC analysis
mentioned in the previous section. Figure 4 depicts a snapshot of the dTEC map at 03:44 UT (00:44 LT) on 18-19 January
2020. The red squares indicate where the TEC disturbance was larger than 0.2 TECu from the running average level. The

spatial resolution of the map, in this case, is around 130 km, and the temporal resolution is 1 min. From previous works
(Figueiredo et al., 2018, Essien et al., 2021), it is known that the amplitude of oscillation of TEC during the passage of MSTIDs
is between 0.1 to 0.2 TECu. Our present results, the dTEC variation of -0.4 to +0.2 TEC are similar to the previous works.


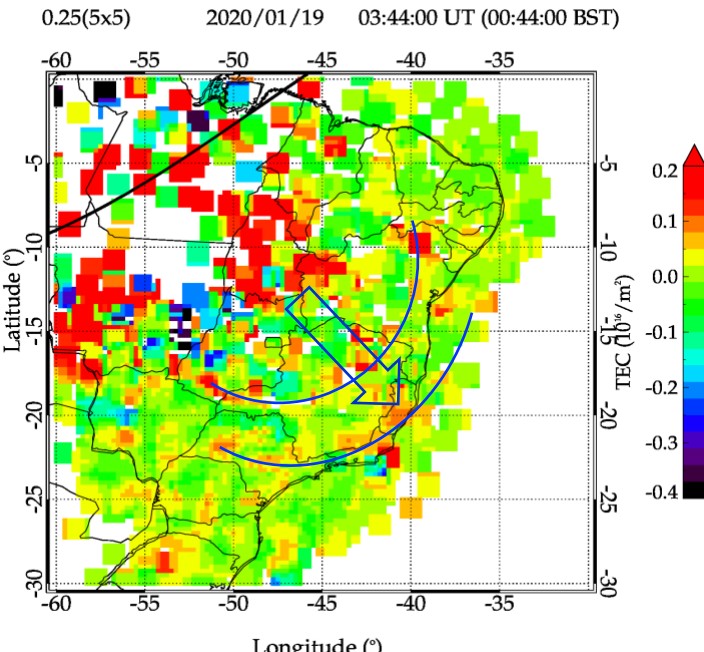

*Fig. 4 dTEC Map in the equatorial and low latitude of Brazil, taken at 03:44 UT (00:44 LT at 45°W), 18-19 January 2020. The black line indicates the geomagnetic equator, and the blue circumferences indicate wave crests of the MSTID. The top/left corner "0.25(5x5)" indicates a spatial resolution of the map as 1.25° x 1.25° (~130 x 130 km).*

The accumulated red squares at around (5- 12° S, 45- 55° W) in Figure 4 indicate a region where the ionosphere is disturbed, most probably due to the EPBs seen in the OI 630 nm images in Figure 3. One can further notice that there is an increase of disturbance (the color of yellow to red squares) on the circumferences indicated by light blue lines, from the East to South region of the map, which is identified as a MSTID-2. The center of the circumferences falls on (12° S, 47° W). Since the wave fronts are curved, it is a concentric wave. The partial concentric form should be caused by the EPB activity in the NW side of the (12° S, 47° W) region where the TEC depletions were going on. It also might be due to the wind filtering effect by the background wind field as reported by Nishioka et al. (2013). From the distance between the wave fronts, the horizontal wavelength was estimated to be 360 ± 50 km. To obtain the wave characteristics, the keogram analysis was also applied. Longitudinally (latitudinally) sliced dTEC map at 13.0° S (43.0° W) as a function of time is shown in Figure 5. During the time interval of 02:30 to 04:00 UT in the 10 to 15° S region (highlighted by the rectangular boxes) one can notice a periodic structure of dTEC perturbation in both longitudinal and latitudinal cuts (indicating by the blue dashed lines), suggesting a passage of disturbance, named as MSTID-2. The disturbance appeared around 01:30 to 04:00 UT in the 47° to 55° W longitudes; it was due to the equatorial plasma bubbles. Although the spatial resolution of the keogram image is not enough to

estimate the horizontal wavelength, the periodicity could be obtained, which is 26.5 ± 2.0 min. The phase velocity is, therefore calculated to be 226 ±15 m/s. The wave characteristics of the MSTID-2 are also summarized in Table 1.


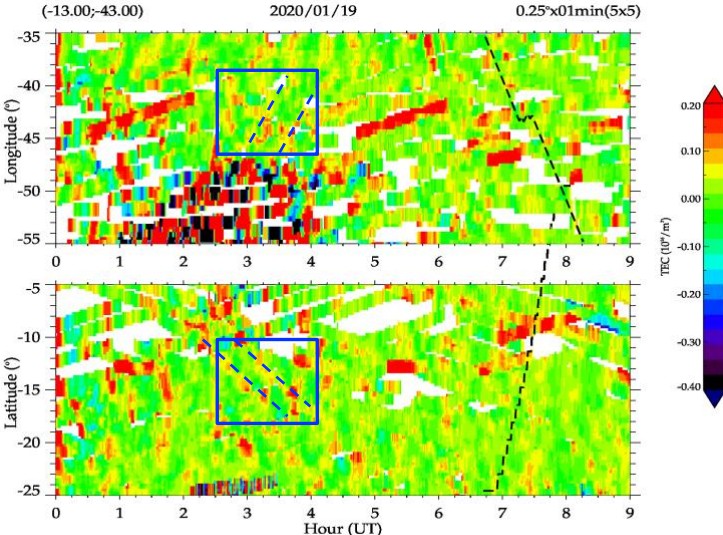

*Fig. 5. Keograms of dTEC: the longitudinal cut at 13.0° S (top) and the latitudinal cut at 43.0° W (bottom) as a function of Time (UT). The rectangular boxes highlight where the FFT spectral analysis was applied.*

**4 Discussion**

Through the evening to midnight over the Bom Jesus da Lapa (BJL) airglow observation site (17° S, 38° W) on 18-19 January 2020, we observed a relatively long wavelength and slow speed GW (GW-2) in the OH emission layer (~87 km altitude) at around 23:00 UT. In 4 hours later (03:00 UT), we observed a wave structure in the OI 630 nm emission layer in the lower ionosphere (~240 km altitude) (MSTID-1). The two different waves, one from MLT and the other from the thermosphere, had almost the same wave characteristics, i. e, a same horizontal wavelength (210 ± 10 km), same period (62 ± 5 min) and the same phase speed (55 ± 5 m/s). The propagation directions of the two emissions, however, are slightly different, OH showing 149° against OI 630 being 113°, the difference of 36°. The OH wavefronts are extended longer than 500 km. On the other hand, the OI 630 wave was limited in the southern sky with a relatively short duration (~60 min). Such coincident occurrence of the wave structure called our attention to further investigate whether these waves have the same origin from the lower atmosphere, *i. e.,* both are primary waves, or one of the waves in the thermosphere was due to the secondary wave generated in the lower atmosphere.

## 4.1 GW Ray-Tracing and Tropospheric Convection Origin

For studying the wave propagation, we used a wave ray-tracing method (Paulino et al. 2013, Vadas et al., 2019, Nyassor et al., 2021) to find out the source of the waves in the lower atmosphere. The wind model used in this work was according to the

NRLMSISE-00 (Picone et al., 2002) and Horizontal Wind Model (HWM14) (Drob et al., 2015). Figure 6 presents the Ray-Tracing trajectories of the GW-2 for the case of no-wind and with-wind model.

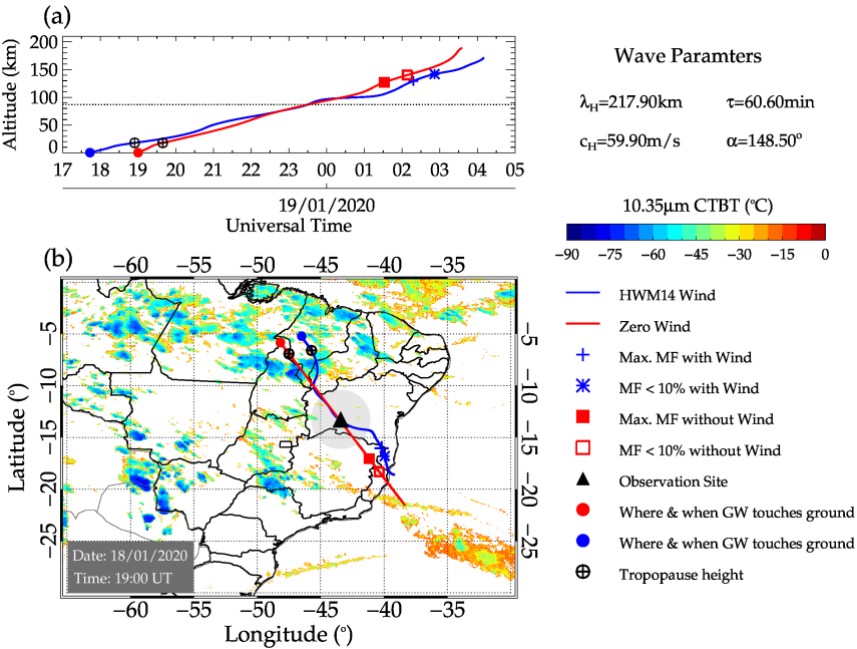

*Fig. 6. Ray tracing (backward and forward) of the observed gravity wave ($\lambda_h$=217.9 km) starting at 87 km altitude at BJL on the night of 18 January 2020 at 23:00 UT. The vertical trajectory versus time (top) and horizontal distance (bottom) are shown.*

*The red line is of the case of no-wind and the blue line is of the with-wind model. The blue triangle is the starting point of the ray tracing. The background map is the cloud top temperature from NOAA GOES 16 meteorological satellite data (10.35 μ) on 18 January 2020 at 19:00 UT.*

The simulation started from 87 km altitude at 23:30 UT and went down (backward tracing) to the ground level at around 19:00

UT, crossing the tropopause (~15 km) at around (7.0° S, 46.5° W) at 19:30 UT. There is a difference of around 200 km of the tropopause crossing positions between the no-wind and with-wind trajectories, which can be assumed to be an error range in the present study. Then, we search for any convective system in this region. Figure 6 also shows the cloud top temperature map produced by GOES-16 (10.35 μ radiation map) at 19:00 UT (https://www.cptec.inpe.br/). It can be seen the convection system spread over the tropical zone 0 -10°S. It is the Intertropical Convergence Zone (ITCZ). One can notice that there is the

lowest temperature spot (-80°C) at (8.5° S, 46.5° W), where a deep convection was in progress. One can notice that this

convection spot is located very close (in an error range of 200 km) to the GW-2 trajectory at the tropopause height. According to the GOES-16 maps, this convection spot started at around 18:00 UT, developing into a much larger area from 19:00 to 23:00 UT and decreasing the intensity after 00:00 UT. During the 5 hours of activity, the convection spot should generate up and down streams inside of the convection cell producing a variety of GWs. The present Ray-Tracing suggests that the observed gravity wave in the OH emission layer started from this convection spot propagating up to the lower thermosphere.


4.2 GW breaking in the thermosphere and generation of a secondary wave

The forward Ray-Tracing shown in Figure 6, on the other hand, went up to 130 km with the momentum flux in the maximum and then it lost the amplitude of oscillation at around 140 km, indicating dissipation of the wave energy. The wave dissipation occurred at the location of (18° S, 40° W) where we observed the wave structure in the OI 630 nm image starting at around 03:00 UT. According to Vadas and Crowly (2010), GW dissipation produces a body force and generates secondary waves. The secondary waves have a variety of wave characteristics. In our present case, we understand that the primary wave observed at the MLT region dissipates in the lower thermosphere, then a secondary wave reaches at the OI 630 emission height, which is located at around 240 km altitude. If this is the case of what happened, the secondary wave had the same characteristic as the primary wave. Vadas and Becker (2018) have discussed in the small- and large-scale secondary waves. According to them, there will be two kinds of secondary waves, one is small scale waves (short horizontal wavelengths) that will be produced during the primary wave breaking process, and the other one is the much longer wavelength (thousands of km) which are produced by a body force generated after the primary GW dissipation. The latter is dependent on the spatial scale of the body force. Bossert et a. (2017), for example, observed secondary waves near the primary (mountain) wave breaking area and found that the horizontal wavelengths are shorter than the primary waves. Smith et al. (2013) reported nearly simultaneous observation of mesospheric GWs by OH airglow and thermospheric GWs by OI 630 nm images. According to their observation, the horizontal wavelength of the OH wave against the OI 630 nm wave is 106 km vs. 255 km, the phase speed of 49.5 m/s vs. 104 m/s, and the period of 36.5 min vs. 42.7 min. Our present case (relatively short wavelength and low phase speed) could be the first case, i.e., secondary wave generated during the primary wave breaking process.

4.3 Possible direct influence of primary wave in the ionosphere

The other possibility of the presence of GWs is a direct influence of the primary wave in the OI 630 nm emission layer. The Ray-Tracing simulation for the MLT GWs did show its dissipation at around 140 km (Figure 6). According to Vadas (2007), the signature of GWs in the thermosphere could be observable even at one or two local density scale heights (15-20 km at around 150 km altitude) above the dissipation altitude. If this is the case, the influence of the primary wave could reach at least at the altitude of 170-180 km where the F-layer bottom side is located. It is worth to check, therefore, the OI 630 nm emission height over BJL during the GW occurrence (03:00-04:00 UT).


The airglow OI 630 nm emission is produced by the dissociative recombination process in the ionosphere:

$O_2^+ + e \rightarrow O(^1D) + O\ (^3P, {}^1S)$

where O($^1$D) is an excited state of atomic oxygen responsible to emit a photon of 630,0 nm. The emission rate, therefore, depends on the concentration of the electron density [e] and its height profile (Chiang et al., 2018). The electron density profile, especially its bottom side profile, could be estimated by ionograms. Unfortunately, there is no ionosonde at BJL. During the passage of the waves at around 03:00 - 03:30 UT (00:00 - 00:30 LT), two DPS ionosondes, one at Fortaleza (7° S, 38° W), the North of BJL, and the other at Cachoeira Paulista (22.7° S, 45.0° W), the South of BJL, were in a routine observation mode. The Fortaleza ionogram showed the F-layer peak height (hmF2) at 220±10 km. It is a mean altitude during the period of 03:00 and 03:20 UT when the ionogram was free from the Spread F condition. It is very low altitude because of the Midnight Collapse of the ionosphere (Gong et al., 2012). On the other hand, the ionosonde at Cachoeira Paulista observed the peak height at 260±10 km. The ionograms used in the present analysis were also manually checked for qualifying the data. From the estimated electron density profiles, we calculated the OI 630 nm volume emission rates based on the equation presented by Chiang et al. (2018). The peak emission altitude at Fortaleza was at 200 km. On the other hand, at Cachoeira Paulista it was at 240 km. The peak altitude at Fortaleza is 40 km lower than Cachoeira Paulista. The BJL site is located between the two ionosonde sites. Therefore, we assume that the OI 630 nm emission peak altitude at BJL might be between 200 and 240 km. In this case, a possibility of that the bottom side of the OI 630 nm emission layer would be disturbed by the primary wave cannot be ruled out. The difference of the propagation direction of 36° between the OH and OI 630 nm wave fronts could be due to the different wind fields between the two emission altitudes.

### 4.4 GWs observed by dTECmap

As shown in Figure 4, we could see a concentric wave structure in the dTEC map. The center of the circular waves is located at around (12.0° S, 47.0° W), which is about the 4.0° South of the convection spot. If one compares the wave characteristics obtained by the dTEC maps (MSTID-2) and the 630 nm images (MSTID-1), the horizontal wavelength of the former is much longer (360 km) and the period is shorter (26.5 min). It means that the phase speed of the former is much faster (226 m/s). Although the characteristics of the two waves are different, they seem to have a same origin. During the period of 19:00 to 23:00 UT, the ITCZ convection spot developed quickly forming a cluster of convection at around (8.0º S, 46.0º W). It could be possible that these convection spots launched a variety of GW spectra. In order to simulate the wave propagation path of the circular fronts, the backward Ray-Tracing was also applied for one of the concentric waves. The wavefront location and its altitude were assumed to be at (17.0º S, 43.0º W) and 300 km altitude, respectively, and started from 03:40 UT. The simulated backward trajectory is shown in Figure 7, where one can see the trajectory went down toward the Northwest and reached the minimum height at around 100 km altitude at (12.5° S, 48.0° W). It means that this concentric wave did not come from the lower atmosphere but was generated above 100 km. Vadas and Azeem (2020) presented (see Figure 21 of Vadas and Azeem (2020)) the backward ray-tracing of the GWs from 300 km altitude for the waves with the different phase speeds, and found that GWs with a phase speeds faster than 200 m/s could not propagate below ~100 km altitude. Our present case is similar to them. The horizontal distance between the wavefront and the location of the minimum height (at 100 km altitude) is approximately 805 km. This location almost coincided with the center of the concentric waves. These facts and

the GW propagation simulation suggest that the concentric waves started above the mesopause and propagated upwards. It indicates that they are not primary waves from the troposphere, but a secondary gravity wave generated in the lower thermosphere.

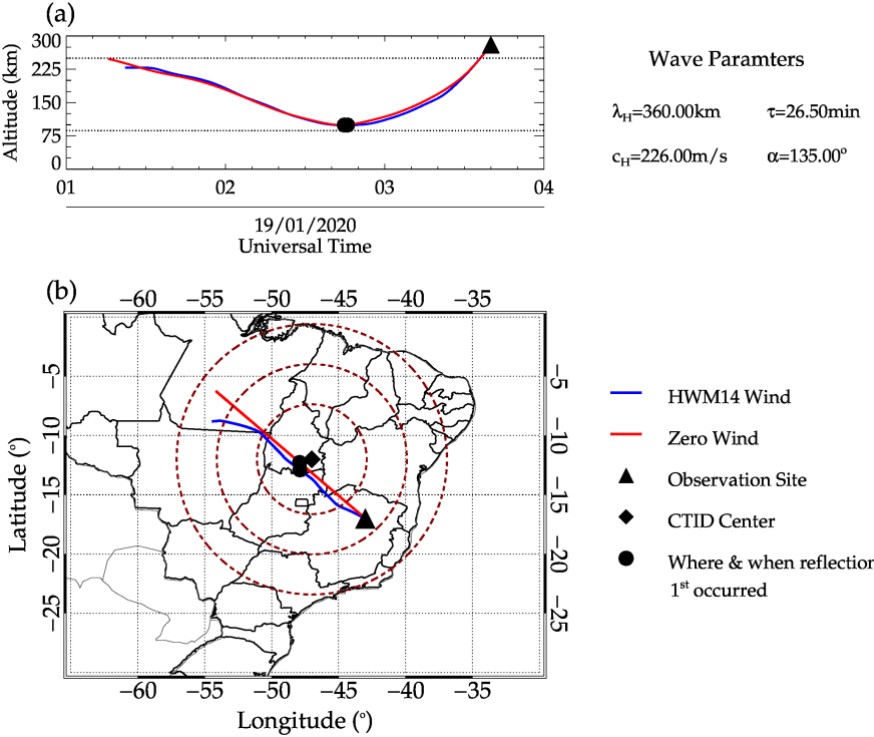

*Fig. 7. Backward Ray-Tracing of the MSTID circular waves, starting from 300 km altitude at (17.0°S, 43.0°W) at 03:40 UT. The vertical trajectory versus time (top) and horizontal distance (bottom) are shown. The red line is in the case of the no-wind and the blue line is of the with-wind model. The black triangle is the starting point of the ray tracing. The black circle at (12.5° S, 48.0°W) indicates where the wave was reflected.*

**Conclusion**


We observed a gravity wave signature at the OH emission height (~ 87 km) and OI 630 nm emission height (< 240 km), those showed the same wave characteristic. Although the two waves look to be similar, the wave observed in the ionosphere might be a secondary wave. However, a direct influence of the primary wave in the OI 630 nm emission layer at around 200 km altitude cannot be ruled out. We also observed fast concentric waves by the dTEC maps, which might be a secondary wave

generated by a primary wave dissipated in the lower thermosphere. Both slow- and high-speed waves have their origin from a convective spot in the ITCZ region. This is the first time to report the direct evidence of GW propagation from the troposphere to the ionosphere by optical imaging and dTEC measurements in the south American region.

**Data availability**

GNSS ground-based receiver data, airglow image data, and ionosonde data used in the present study are available at the
EMBRACE data center website (http://www2.inpe.br/climaespacial/portal/ en/#). The satellite infrared thermal images (Figure 7) are obtained from the Geostationary Operational Environmental Satellite System 16 (GOES 16) data (http://satelite.cptec.inpe.br/home/ index.jsp), provided by the Center for Weather Forecasting and Climate Studies (CPTEC) in Brazil. Two atmospheric models were used in computing the gravity wave ray tracing (Figure 6): one is the MSIS-E-00 Atmospheric Model: https://ccmc.gsfc.nasa.gov/modelweb/models/msis_vitmo.php and the other is the empirical Horizontal
Wind Model (HWM14) (Drob et al., 2015).

**Competing interests:**

The authors have no competing interests with any other groups.

**Author Contribution:**

Hisao Takahashi: Data analysis and interpretation
Cosme A. O. B. Figueiredo: Data analysis
Patrick Essien: Data analysis
Cristiano M. Wrasse: Data interpretation
Diego Barros: Data analysis
Prosper K. Nyassor: Data analysis
Igo Paulino: data analysis
Fabio Egito: data handling
Geangelo M. Rosa: data handling
Antonio H. R. Sampaio: data handling

**Acknowledgements**

The present work was supported by CNPq (Conselho Nacional de Pesquisa e desenvolvimento) under the grants 310927/2020-0, 150569/2017-3, 161894/2015-1, 303511/2016, 300322/2022-4 and 306063/2020-4; Fundação de Amparo à Pesquisa do Estado de São Paulo (FAPESP) under the grant 2018/09066-8 and 2019/22548-4; and Coordenação de Aperfeiçoamento de Pessoal de Nível Superior (CAPES) under the process BEX4488/14-8. Paulino thanks to Fundação de Amparo à Pesquisa do Estado da Paraíba for the grants Demanda Universal Edital 09/20221 and Edital PRONEX termo de concessão 002/2019.

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

## Table Caption

***Table 1.*** *Wave Characteristics obtained by the OH images in the MLT region (GW-1 and GW-2), and OI 630 nm images in the thermosphere (MSTID-1) and dTEC maps in the ionosphere (MSTID-2). (The values in parentheses are the error range.)*

## Figure Captions

*Fig. 1. Geographically coordinated OH images observed at Bom Jesus da Lapa (BJL), 13.3°S, 43.5°W, geomag.14.1°S, at 23:30 UT (left), 23:41 UT (center) and 23:51 UT (right) on the night of 18-19 January 2020. Blue dots indicate the zenith of BJL. The blue arrows indicate the wave fronts of the longer wavelength one (GW-2). The red-arrow indicates the direction of propagation.*

*Fig. 2. Keograms of OH images observed at BJL in the night of 18-19 January 2020. Meridian cuts along the W-*
*E direction (upper panel) and the S-N direction (lower panel) are shown as a function of time from 22:00 to 30:00 (06:00) UT. The red color rectangular boxes are where the FFT spectral analysis were taken.*

*Fig. 3. Geographically coordinated OI 630 nm images observed at Bom Jesus da Lapa (BJL) at 03:20 UT (left), 03:34 UT(center) and 03:48 UT (right) on the night of 18-19 January 2020. The top 3 images are original and the bottom 3 images are residual subtracted from the one hour averaged image. Blue dots indicate the location of BJL.*
*MSTID-1 (lower/middle), EPBs (upper left corner) and MTM (upper left) can be seen (see Text).*

*Fig. 4 dTEC Map in the equatorial and low latitude of Brazil, taken at 03:44 UT (00:44 LT at 45°W), 18-19 January 2020. The black line indicates the geomagnetic equator, and the blue circumferences indicate wave crests of the MSTID. The top/left corner "0.25(5x5)" indicates a spatial resolution of the map as 1.25° x 1.25°(~130 x 130 km).*

*Fig. 5. Keograms of dTEC: the longitudinal cut at 13.0°S (top) and the latitudinal cut at 43.0°W (bottom) as a*
*function of Time (UT). The rectangular boxes highlight where the FFT spectral analysis was applied.*

*Fig. 6. Ray tracing (backward and forward) of the observed gravity wave ($\lambda_h$=217.9 km) starting at 87 km altitude at BJL on the night of 18 January 2020 at 23:00 UT. The vertical trajectory versus time (top) and horizontal distance (bottom) are shown. The red line is of the case of no-wind and the blue line is of the with-wind model. The*

*blue triangle is the starting point of the ray tracing. The background map is the cloud top temperature from NOAA*

*GOES 16 meteorological satellite data (10.35 μ) on 18 January 2020 at 19:00 UT.*

*Fig. 7. Backward Ray-Tracing of the MSTID circular waves, starting from 300 km altitude at (17.0°S, 43.0°W) at 03:40 UT. The vertical trajectory versus time (top) and horizontal distance (bottom) are shown. The red line is in the case of the no-wind and the blue line is of the with-wind model. The black triangle is the starting point of the ray tracing. The black circle at (12.5°S, 48.0°W) indicates where the wave was reflected.*
