# Peer review of "Signature of gravity wave propagations from the Troposphere to Ionosphere"

_Annales Geophysicae, 2022_

## Referee Comment (RC2)

Comments on, **"Signature of gravity wave propagations from the Troposphere to Ionosphere"** by Takahashi et al.

Using an airglow imaging observation, the authors observed gravity waves (GWs) in the mesosphere and MSTIDs in the thermosphere 4 hours later. Concurrent with the MSTIDs in the airglow images, concentric waves were also observed in the GPS detrended TEC. Furthermore, using the raytracing technique, source and dissipation region of the GWs is identified. From the obtained results, they argue that the source and dissipation region of the GWs are troposphere and lower thermosphere, respectively. The authors also postulate that the concentric wave could be a secondary wave caused by the breaking of the primary waves in the lower thermosphere. The objective of the manuscript is interesting, however, the results and discussion are not convincing. Though they have the seed but it is not enough to publish in the present form. For example, the authors mentioned that there is a MSTIDs in the 630 nm images and concentric waves in the dTEC but it is not clear in the Figures 3 and 4. I believe that in the dTEC, the western part bright patches are due to equatorial plasma bubbles (EPB) not due to the concentric waves, if the authors remove the artificial circles from the map we cannot see any concentric waves. Similarly, I could see 4 wave fronts in the OH images but the authors emphasizes only two, why? It is better to attach the movies of the images as a supplementary document or provide more images (dTEC maps) to show the evolution of the waves and MSTIDs. Therefore, I recommend to the editor for a major revision. The line by line comments are as follows:

**Major comments:**

1. In addition to the actual images, include the residual images to represent the waves and MSTIDs in a better way. Similarly, instead of showing only one or two images, the authors should show few more images in Figures 1, 3, and 4. Otherwise, it is hard to believe the existence of waves exist in the images? Particularly, in figures 3 and 4, I could not see the MSTIDs and concentric waves. Additionally, include the movies of the images and dTEC maps for this night in the supporting information.
2. Considering the number of the wave fronts (the area separation between them) and area of the OH images, I strongly believe that the horizontal wavelength of the GWs shown in this study does not exceed even 100 km. Check this carefully and comment on it. The residual images will represent the GWs and MSTIDs in a better way than the actual images.
3. By comparing the OI630 nm images and dTEC maps, one can understand that the red patches around the region of -45° to -60° longitude and -15° to 0° latitude is due to EPB. Moreover, if the authors remove the artificial circle, the concentric circle cannot be seen. The figure presented in the manuscript is not convincing. So, it is important to show the consecutive dTEC maps or the movie of dTEC maps.
4. Why the MSTIDs are not seen in the dTEC maps? Comment on it.
5. The red and blue bands in the keograms are most probably due to the EPB because the latitude considered for EW keogram is close to the equator, more importantly, the longitude (latitude in the NS keogram) where the bands are noted is exactly same where the EPB are observed in the airglow images. Justify this.

**Minor comments:**

6. From figures 1 and 2, one can see at least 4 clear wave fronts but why the authors emphasize only 2? In figure 2, NS keogram between 100- 200 km (~22-25 UT) there were four wave fronts but why it is not highlighted?
7. Line 130, how is the FFT analysis carried out?
8. Line 225 between 3-4 UT at least 4 data points should be available from the ionosonde observations, is the 220 km hourly mean value?
9. We could see the EPB in the images, during this condition how much reliable are the ionosonde hmF2 values?

---

## Author Comment (AC2)

Review Report of Reviewer #2 for the manuscript `angeo-2022-13`

(Please note that the text with **blue fonts** is Author's Reply and red fonts are a copy of the revised manuscript).

Reply to Reviewer 2

Using an airglow imaging observation, the authors observed gravity waves (GWs) in the mesosphere and MSTIDs in the thermosphere 4 hours later. Concurrent with the MSTIDs in the airglow images, concentric waves were also observed in the GPS detrended TEC. Furthermore, using the raytracing technique, source and dissipation region of the GWs is identified. From the obtained results, they argue that the source and dissipation region of the GWs are troposphere and lower thermosphere, respectively. The authors also postulate that the concentric wave could be a secondary wave caused by the breaking of the primary waves in the lower thermosphere. The objective of the manuscript is interesting, however, the results and discussion are not convincing. Though they have the seed but it is not enough to publish in the present form. For example, the authors mentioned that there is a MSTIDs in the 630 nm images and concentric waves in the dTEC but it is not clear in the Figures 3 and 4. I believe that in the dTEC, the western part bright patches are due to equatorial plasma bubbles (EPB) not due to the concentric waves, if the authors remove the artificial circles from the map we cannot see any concentric waves. Similarly, I could see 4 wave fronts in the OH images but the authors emphasizes only two, why? It is better to attach the movies of the images as a supplementary document or provide more images (dTEC maps) to show the evolution of the waves and MSTIDs. Therefore, I recommend to the editor for a major revision. The line by line comments are as follows:

Reply: We thank for critical comments given by the Reviewer. We agree with his (her) comments. In order to make it clear to show the wave structure of the OI 630 nm images and dTEC maps, Figure 1 (OH images), Figure 3 (630 images), Figure 4 (dTEC map) and Figure 5 (dTEC keogram) were revised. Also 630 movie is attached for supplement. Hope that this revised version will attend the reviewer's suggestions and to improve what we want to present.

Major comments:

1. In addition to the actual images, include the residual images to represent the waves and MSTIDs in a better way. Similarly, instead of showing only one or two images, the authors should show few more images in **Figures 1, 3, and 4.** Otherwise, it is hard to believe the existence of waves exist in the images? Particularly, in figures 3 and 4, I could not see the MSTIDs and concentric waves. Additionally, include the movies of the images and dTEC maps for this night in the supporting information.

Reply: According to the reviewer's suggestion, we revised:

Figure 1 (OH images): 3 consecutive images of the OH emission, where one can clearly see the wave front displacement. The blue arrows are added to show where the long wave (GW-1) fronts are located.

Figure 3 (OI 630 nm images): According to the reviewer's suggestion, 3 original images (a, b, c) and 3 residual images ([Image]$_t$ – [one hour averaged image]) are shown (d, e, f). As the Reviewer pointed out that the dark bands of the top left corner (the direction of NW) are the equatorial plasma bubbles, and the two extended dark bands from the bottom (South) are the MSTIDs. The difference between the plasma bubble and MSTID is clear to see (the MSTIDs are located in the southern sky not reaching the equator). The text was revised as follows:

(Line 135) The difference between the EPBs and MSTID-1 is clear to see. The EPBs are extending from the equator side and the MSTID is elongating from the south. During the 28 min of the time interval (from Figure 3(a) to (c)), a dark band moved toward East by ~90 km.

Figure 4 (dTECMap): We could not find a sequence of dTECmaps with a good quality of wave structure during the period of 03:00 to 04:00, except the map at around 03:44 UT. The reason was due to the spatially low resolution (~130 km wide) and the small amplitude of variation (dTEC< 0.1). This is why we used keogram to find out the propagation mode and to estimate the period of oscillation (Figure 5). Therefore, we decided to keep only one dTEC map at 03:44 UT.

Further regarding Figure 4, we re-drawn the concentric circumferences instead of the circles we used previously. The Reviewer had a reason to point out. As mentioned in the text, the west and north side of the map did show larger TEC variations due to the plasma bubble activity. The text was revised as follows:

(Line 171) The accumulated red squares at around (5-12°S, 45-55°W) in Figure 4 indicate a region where the ionosphere is disturbed, most probably due to the EPBs seen in the OI 630 nm images in Figure 3.

Movies for supporting files: we included a movie for the OI 630 nm image. Concerning a movie for dTECmap, we decided not to present, because of the low resolution of the maps and it is difficult to show concentric form of the MSTID wave structure.

2. Considering the number of the wave fronts (the area separation between them) and area of the OH images, I strongly believe that the horizontal wavelength of the GWs shown in this study does not exceed even 100 km. Check this carefully and comment on it. The residual images will represent the GWs and MSTIDs in a better way than the actual images.

Reply: Regarding the OH images (Figure 1), we can see two long wave fronts and two short wave fronts in the OH image in Figure 1(a). We are sorry for the lack of explanation it in the text. One of the long wave fronts disappeared in the SE side in the Figure 1(b).

[Figure]

For better presenting the two long wave fronts in the Figure 1, blue arrows are added in the figure for reference. In the figure, we can see another short wavelength (36 km), now it is named as GW-1, together with the long one (218 km), now it is called as GW-2. In the OH image keogram (Figure 2), one can clearly recognize the long wave propagation. We revised Figure 1 and **Table 1** including the short wave (GW-1) characteristics, too.

The text was revised as fllows:

(Line 98) In Figure 1(a), there are two wave structures, one is shorter wavelength in the NW side of the image (top left side) (GW-1) and the another is a longer wavelength, one wavefront over the zenith (blue dot) and the other at SE, indicated by blue arrows (GW-2).

3. By comparing the OI630 nm images and dTEC maps, one can understand that the red patches around the region of -45° to -60° longitude and -15° to 0° latitude is due to EPB. Moreover, if the authors remove the artificial circle, the concentric circle cannot be seen. The figure presented in the manuscript is not convincing. So, it is important to show the consecutive dTEC maps or the movie of dTEC maps.

Reply: As the reviewer pointed out, the red patches around the region of -45 to -60 longitude and 0 to -14 latitude are due to the disturbance caused by EPBs. The concentric MSTID started at around (-12, -47) propagating toward the southeast. The EPBs and MSTIDs are not overlapping. Regarding the concentric form of MSTID in Figure 4, the wave fronts can be seen partially in a region of the southeast. It is a curved form, extending by a quarter circle. The full circles drawn in the figure in the original version could mislead readers. Therefore, we re-drawn them by circumferences. We explained it in the text why the concentric wave was partially formed.

(Lines 174-) Since the wave fronts are curved, it is a concentric wave. The partial concentric form could be caused by the EPB activity in the NW side of (12°S, 47°W) where the TEC depletions were going on. It also might be due to the wind filtering effect by the background wind field as reported by Nishioka et al. (2013).

[Figure]

Regarding the Reviewer's suggestion to show the consecutive dTECmaps or the movie of dTECmaps, we decided not to present them. Because of the low amplitude of oscillation (< 0.1 TECu) and low spatial resolution (>130 km), it was difficult to produce a good quality of dTECmap. The dTECmap at 03:44 UT (Figure 4) is one of the best cases. This is the reason to use keogram to find out any systematic wave front movement (Fig. 5).

4. Why the MSTIDs are not seen in the dTEC maps? Comment on it.

Reply: The MSTID observed by the OI 630 nm image has the horizontal wavelength of 218 km. The horizontal resolution of the dTEC map, on the other hand, is > 130 km. So, it is difficult to identify it. The longer MSTID wave (360 km), on the other hand, is able to see in the dTECmap.

5. The red and blue bands in the keograms are most probably due to the EPB because the latitude considered for EW keogram is close to the equator, more importantly, the longitude (latitude in the NS keogram) where the bands are noted is exactly same where the EPB are observed in the airglow images. Justify this.

Reply: The Reviewer has a reason. In the previous version, we presented the keograms of the region where the EPBs are moving, and could not to see signature of MSTIDs. We are sorry for that. In the revised version Figure 5 (below), we selected the latitudinal cut from -5 to -25 at 43W longitude, and the longitudinal cut at 13S latitude. One can see two extended disturbances starting from the (12S, 45W) region, which is the MSTID. The text was revised as:

(Lines 178) To obtain the wave characteristics, the keogram analysis was also applied. Longitudinally (latitudinally) sliced dTECmap at 13.0°S (43.0°W) as a function of time is shown in Figure 5. During the time interval of 02:30 to 04:00 UT in the 10° to 15°S region (highlighted by rectangular boxes) one can notice a periodic structure of dTEC perturbation in both longitudinal and latitudinal cuts (indicating by the blue dashed lines), suggesting a passage of MSTID.

[Figure]

Minor comments:

6. From figures 1 and 2, one can see at least 4 clear wave fronts but why the authors emphasize only 2? In figure 2, NS keogram between 100- 200 km (~22-25 UT) there were four wave fronts but why it is not highlighted?

Reply: The reviewer has a reason. In Figure 1(OH images) and Figure 2 (keogram), there are 4 wave fronts, two are brighter and longer, and the other two fronts are shorter ones. In the text we only selected the longer ones and did not comment the shorter ones. Sorry for that. In the revised version, therefore, we present the two different waves, and pointing the longer ones by arrows in Figure 1. The characteristics of the short wave are now included in the revised Table 1. The text was revised as follows:

(Line 98) In Figure 1(a), there are two wave structures, one is shorter wavelength in the NW side of the image (top left side) (GW-1) and the another is a longer wavelength, one wavefront over the zenith (blue dot) and the other at SE, indicated by blue arrows (GW-2).

7. Line 130, how is the FFT analysis carried out?

Reply: We used the FFT method to find out wave characteristics, horizontal wavelength, period, and the phase speed from the keogram. The detail of calculating procedure did not present here because of that it has been presented by Wrasse et al. (2007) and Figueiredo et al, JGR (2018). The use of FFT and related references are already mentioned in the previous paragraph of OH image analysis (Line 121). Therefore, we revised the text in the following:

(Line 138) In order to get the wave characteristics of the OI 630 images, we used the FFT spectral analysis for the OI 630 keogram (not shown here), which is similar to the OH image analysis mentioned above (Wrasse et al, 2007).

8. Line 225 between 3-4 UT at least 4 data points should be available from the ionosonde

observations, is the 220 km hourly mean value?

Reply: The hmF2 value (220 km) at Fortaleza is a mean of 3 points from 03:00 to 03:20 UT. The same procedure was taken for ionosonde data at Cachoeira Paulista. In order to expain it, the text was revised as follows:

(Line 259) The Fortaleza ionogram showed the F-layer peak height (hmF2) at 220±10 km. It is a mean altitude during the period of 03:00 and 03:20 UT ,,,,

9. We could see the EPB in the images, during this condition how much reliable are the

ionosonde hmF2 values?

Reply: We used the ionogram data (hmF2) at Fortaleza (FZ) and Cachoeira Paulista (CP)when the Spread F was not over zenith and the ionogram traces could be seen clearly. In order to clarify it, we revised the text:

(Line 259) The Fortaleza ionogram showed the F-layer peak height (hmF2) at 220±10 km. It is a mean altitude during the period of 03:00 and 03:20 UT when the ionogram was free from the Spread F

---

## Author Comment (AC3)

RC1: 'Comment on angeo-2022-13', Anonymous Referee #1, 14 Jun 2022  reply

(Please note that the text with **blue fonts** is Author's Reply and red fonts are a copy of the revised manuscript).

Comments: This is interesting study which brings some new information but which requires some improvements before being published.

Concentric waves in dTEC are secondary GWs as I would expect. TEC is integral parameter; nevertheless the secondary GWs could hardly produce quite different wavy oscillations in TEC and in OI 630 nm emission from heights relatively close to the F-region maximum. Therefore possibility that the effect in OI 630 nm, which is essentially the same as the effect of primary waves in OH emission in the mesopause region, is caused by primary GWs unable to propagate well above the OI 630 nm height and affect TEC, seems to me to be probable.

Reply: We thank for the Reviewer's comment that the effect in OI 630 nm is caused by primary GWs, and that of the dTEC are due to secondary waves. We understand in his (her) points of view. Since there is no definitive proves to conclude it, we discussed in the two possibilities, primary and secondary GWs. On the present stage, we prefer to summarize that there are two possible explanations.

Section 2, Observations: I recommend add the analyzed period.

Reply: Yes, the reviewer has a reason. We included the following sentences in the Section of "Observation"

(Line 75): In the present study we used the image data from December 2019 to September 2020. During this period, we selected 13 days to analyze wave structures in the OI 630 nm images.

Line 173: the directions of propagation of GWs in OH and 630 nm emissions are relatively large, they cannot be considered to be almost same.

Reply: We agree with the reviewer's comment. Regarding the direction of propagation, we included the following sentence:

(Line 197) The propagation directions of the two emissions, however, are slightly different, OH showing 149° against OI 630 being 113°, the difference of 36°.

And the difference of 36 deg is commented at the section of 4.3 Possible direct influence of primary wave in the ionosphere:

 (Line 268) The difference of the propagation direction of  36° between the OH and OI 630 nm wave fronts could be due to the wind fields between the two emission altitudes.

Wording and misprints:

Line 44: "et al," should be "et al." Yes, corrected.

Line 50: "et al.," should be "et al." Yes, corrected.

Line 57: "generating" should be "generates." Yes, corrected.

Line 61: "reach the mesosphere to the lower thermosphere". What do you mean, from troposphere to MLT or from mesosphere to the lower thermosphere (I expect the former).

Reply: We agree with the reviewer's comment, and revised the sentence as follows:

(Line 62): It would take several hours to reach from the troposphere to the mesosphere-lower thermosphere (Vadas and Liu, 2013), which makes it difficult to follow the wave step by step.

Line 84: "one TEC" should be "one TECU"; similar corrections throughout the paper.

Reply: Yes, we revised "one TEC" to "one TECu".

Line 98: "an red" should be "a red": Yes, corrected.

Line 126: "could not observe" should be "could be observed": Yes, corrected.

Line 160: "region was" should be "region; it was": Yes, corrected.

---

## Author Response (AR2)

ANGEO-2022-13 | Regular paper

Signature of gravity wave propagations from the Troposphere to Ionosphere, by Takahashi et al.

Reply to Reviewer #2

*(Comments):The revised manuscript has improved significantly. However, the dTEC part is not convincing because it is hard to see any concentric wave structure in the dTEC as well as in the keograms. Thus, concentric waves in the dTEC figures should be shown convincingly or dropped from the manuscript. I believe without that portion also the manuscript has significant results.*

Reply: It is pity that the reviewer did not accept our interpretation of the observed dTEC wave structure (Figure 4 in the 2$^{nd}$ version). On the other hand, we understand in the doubt pointed out by the reviewer (not clear wave fronts in the figure). This is due to the low spatial resolution of the dTEC map. Therefore, we accept the reviewer's argument, and the results and discussion related to the dTEC map (Figures 4, 5, 7) were deleted in the text.

Please note that the conclusion of the present work had no change by this modification.

*(Comment):In the introduction section, studies using the airglow imager from the Asian sector are missing (e.g. Japan and India sectors, there are few papers published with the same objectives).*

Reply: Yes, we agree with the Reviewer's comment. Three references were included in the "Introduction", (line 37):

Dynamical processes in the mesosphere to thermosphere were studied by OH and oxygen 630.0 nm airglow imaging by Kubota et al. (2000), Taori et al. (2013) and most recent by Ramkumar et al. (2021).

We thank for reviewer's critical discussion and constructive suggestion of the Editor,

Hisao Takahashi (corresponding author)